# Sexual Dimorphism in Kisspeptin Signaling

**DOI:** 10.3390/cells11071146

**Published:** 2022-03-28

**Authors:** Eun Bee Lee, Iman Dilower, Courtney A. Marsh, Michael W. Wolfe, Saeed Masumi, Sameer Upadhyaya, Mohammad A. Karim Rumi

**Affiliations:** 1Department of Pathology and Laboratory Medicine, University of Kansas Medical Center, Kansas City, KS 66160, USA; elee10@kumc.edu (E.B.L.); idilower@kumc.edu (I.D.); s576m282@kumc.edu (S.M.); supadhyaya@kumc.edu (S.U.); 2Department of Molecular and Integrative Physiology, University of Kansas Medical Center, Kansas City, KS 66160, USA; cmarsh2@kumc.edu (C.A.M.); mwolfe2@kumc.edu (M.W.W.); 3Department of Obstetrics and Gynecology, University of Kansas Medical Center, Kansas City, KS 66160, USA

**Keywords:** kisspeptins, kisspeptin neurons, GnRH neurons, gonadotropins, extrahypothalamic kisspeptins, sexual dimorphism

## Abstract

Kisspeptin (KP) and kisspeptin receptor (KPR) are essential for the onset of puberty, development of gonads, and maintenance of gonadal function in both males and females. Hypothalamic KPs and KPR display a high degree of sexual dimorphism in expression and function. KPs act on KPR in gonadotropin releasing hormone (GnRH) neurons and induce distinct patterns of GnRH secretion in males and females. GnRH acts on the anterior pituitary to secrete gonadotropins, which are required for steroidogenesis and gametogenesis in testes and ovaries. Gonadal steroid hormones in turn regulate the KP neurons. Gonadal hormones inhibit the KP neurons within the arcuate nucleus and generate pulsatile GnRH mediated gonadotropin (GPN) secretion in both sexes. However, the numbers of KP neurons in the anteroventral periventricular nucleus and preoptic area are greater in females, which release a large amount of KPs in response to a high estrogen level and induce the preovulatory GPN surge. In addition to the hypothalamus, KPs and KPR are also expressed in various extrahypothalamic tissues including the liver, pancreas, fat, and gonads. There is a remarkable difference in circulating KP levels between males and females. An increased level of KPs in females can be linked to increased numbers of KP neurons in female hypothalamus and more KP production in the ovaries and adipose tissues. Although the sexually dimorphic features are well characterized for hypothalamic KPs, very little is known about the extrahypothalamic KPs. This review article summarizes current knowledge regarding the sexual dimorphism in hypothalamic as well as extrahypothalamic KP and KPR system in primates and rodents.

## 1. Kisspeptins and Kisspeptin Receptors

### 1.1. Kisspeptin Signaling

Kisspeptins (KPs) are neuropeptides encoded by the *Kiss1* gene in hypothalamic KP neurons as well as in several extrahypothalamic cell types. KPs are expressed as a 145 amino acid peptide, which is initially processed to 54 amino acids in primates, and 52 amino acids in rodents [1]. The 54/52 amino acid peptide is further processed to 14, 13, or 10 amino acids peptides. The carboxy-terminal portion of KP54/52 reported to be as potent as KP-54/52 [2] (Figure 1). The *Kiss1r* (*Gpr54*) gene encodes for the KP receptor (KPR), which is a G-protein coupled receptor consisting of 398 amino acids [3]. KPs are RF-amide (R-F-NH2) type of neuropeptides in primates but in the rodents, KPs are RG-amides (R-G-NH2) [4]. Binding of KPs to KPR activates the receptor coupled Gαq/11 and the G-protein-activated phospholipase C that induce second messengers, inositol triphosphate (IP3) and diacylglycerol (DAG) [4]. While IP3 mediates the intracellular Ca^2+^ release, DAG activates the PKC pathway [4]. KP stimulation also activates the MAP kinase ERK1/2 [5]. However, the downstream molecular consequence of KP-signaling varies among the cell types in different tissues that express the KPR [4]. 

The primary role of hypothalamic KPs is the regulation of gonadotropin releasing hormone (GnRH) secretion from GnRH neurons [6,7]. GnRH acts on the gonadotrophs in the anterior pituitary gland to release gonadotropins (GPNs)- namely follicle stimulating hormone (FSH), and luteinizing hormone (LH) [6]. KPs have been shown to act directly on pituitary gonadotrophs in many species to induce GPN secretion [8]. GPNs are essential for gonadal development as well as gonadal functions in both sexes [8]. KPs also regulate the onset of puberty, feedback responses to gonadal steroid hormones, and induce the preovulatory GPN surge [4]. Thus, KP-signaling is an essential regulator of the hypothalamic-pituitary-gonadal (HPG) axis. Recent studies suggest that KP signaling also serves numerous regulatory functions other than GnRH secretion including metabolism, placentation, and emotions [4,9,10].

### 1.2. Hypothalamic Kisspeptins and Kisspeptin Receptors

Gonadal steroid hormones are produced in response to GPNs secreted from the hypothalamic-pituitary (HP) axis and the gonadal hormones in turn regulate GPN secretion [11,12]. However, the neurons that produce GnRH to regulate the pituitary secretion of GPNs, lack receptors for gonadal steroid hormones [13]. KPs and KPR bridge the connection between the gonadal hormones and GnRH neurons in hypothalamus, which is their most important and well-recognized physiological function [14].

Hypothalamic KPs regulate pituitary GPN secretion in mammals, including rodents, sheep, and primates [8,15,16,17]. KP neurons are present in the hypothalamic arcuate (ARC) nuclei and infundibular region of rodents and primates [8,17]. A second group of KP neurons is found in the anteroventral periventricular (AVPV) nuclei in rodents and the periventricular preoptic nucleus (PeN) in sheep and primates [8,17,18]. KP neurons in the ARC nuclei mediate the negative feedback of sex steroid hormones and induce pulsatile GnRH secretion [8]. Whereas KP neurons in AVPV/PeN nuclei generate the preovulatory surge of GnRH and GPNs in response to a high level of estrogen [8]. The KPR is expressed in very high levels in hypothalamic GnRH neurons, which suggests the importance of KP-signaling in these cells [19]. Inactivating mutations of KP or KPR gene lead to hypogonadotropic hypogonadism in men and women [20,21]. Loss of KP-signaling in KP^KO^ or KPR^KO^ mice also results in infertility indicating it is essential for mammalian reproduction [22,23,24,25]. GnRH-specific KPR^KO^ mice were also found to be infertile [26]. Moreover, the reproductive phenotype of global KPR^KO^ mutant mice was rescued by selective expression of KPR in the GnRH neurons, suggesting that KP-signaling in hypothalamic GnRH neurons is sufficient for maintaining reproductive function [27]. However, recent studies also emphasize the importance of KP-signaling within the male and female gonads [28,29,30].

### 1.3. Extrahypothalamic Kisspeptins and Kisspeptin Receptors

The role of KP and KPR at the hypothalamic-pituitary level is well known [31]. However, recent studies suggest that the role of KP-signaling may not be limited to the neuroendocrine regulation of reproductive functions. KPs and KPR are also expressed in many extrahypothalamic tissues inside and outside of the brain. In both rodents and primates, KP-expressing cells are detected in the medial nucleus of the amygdala, as well as in the bed nucleus of stria terminalis [32]. KP and KPR have also been detected within the pituitary gland of rodents and primates and KP-signaling can induce the expression of GPNs [8]. Numerous studies have shown that KPs and KPR are expressed in peripheral organs including liver [33,34], pancreas [34], fat [34,35,36], adrenal gland [37], heart [38], testis [34,39,40], ovary [28,41,42,43], uterus, and placenta [44,45,46]. Among the potential functions of extrahypothalamic KP-signaling, the most studied are related to reproductive functions of gonadal KPs and regulation of metabolism by KPs in the liver, pancreas, and adipose tissues [28,34,39,40,41,42,43,45]. Nevertheless, the precise role of extrahypothalamic KPs remains largely unclear.

### 1.4. Circulating Kisspeptins

Circulating KPs levels in children are higher than that of adults [47]. Even children under 9 years of age have 3-fold higher levels of KPs than adults, which is further increased at the 9 to 12 years of age [48] (Figure 2). There is no significant difference in plasma levels of KP between males and females during early puberty [49,50]. Circulating KP levels and urinary KPs are similar among boys and girls under 12 years; however, there is a sex-specific difference thereafter [51]. After puberty, KP levels increase in women compared to that of men [52]. However, both adult males and females exhibit decreasing levels of KPs with increasing age [53].

This difference in circulating KP levels between men and women suggests that there is sexual dimorphism in regulation of KP expression [53]. Circulating KP originates from either hypothalamic or extrahypothalamic sources [54]. Sexual dimorphism in circulating KPs may also be related to a higher number of KP neurons in females, as well as increased KP synthesis in the ovary and adipose tissue [53]. It may also be related to hormonal changes as sex hormones have been shown to influence KP levels. Studies have demonstrated that the liver and placenta can contribute a significant amount of KPs to circulating levels [33,44,54]. Circulating KP levels increase ~7000-fold during the pregnancy [44]. This increase in KP levels may contribute to hormonal and metabolic adaptations during pregnancy, however the significance of such incredible rise in KP levels remains unknown [8]. Cancers involving the extrahypothalamic tissues have also been associated with elevated levels of KP in circulation [55]. Nevertheless, it remains unclear whether the circulating KPs from peripheral sources impacts the HPG axis and reproductive function.

### 1.5. Coexpression of Neuropeptides with Kisspeptins

KP neurons within the ARC nucleus coexpress neurokinin B and dynorphin and have been identified as KNDy neurons [56]. KNDy neurons are the key regulators of pulsatile GnRH [56], and GPN secretion [57,58,59]. They express both androgen and estrogen receptors and relay negative feedback by steroids to alter GnRH neuron activity [60,61,62,63]. KPs are strong activators of GnRH neurons, while NKB and dynorphin act as intrinsic modulators of KP release from KP neurons [64]. KP neurons also express the classical neurotransmitters; GABA in AVPV/PeN nuclei and glutamate in ARC nuclei [65]. Consequently, KP neurons in AVPV/PeN nuclei inhibit while those in ARC nuclei stimulate the neurons that control food intake and energy expenditure [65]. Studies have shown that a subpopulation of KP neurons in AVPV/PeN area also express GABA [66,67,68]. It is suspected that both KPs and GABA can regulate the activity of GnRH neurons [69,70,71], however, the exact role of GABA-signaling in the induction of GPN surge remains undetermined [72].

### 1.6. Regulation of Hypothalamic and Extrahypothalamic Kisspeptin Expression

Sex differences in steroid hormone receptor expression in KP neurons has been suggested to impact the sexually dimorphic LH surge in mice [73]. The KP neurons in ARC and AVPV nuclei express both androgen receptor and estrogen receptor α (ERα) [60,68,74]. Testosterone negatively regulates the ARC nuclei and positively regulates the AVPV nuclei in males, which is similar to estrogen mediated regulation of the KP neurons in females [60,74]. It has been reported that the effects of testosterone can be mediated by either androgen receptor or estrogen receptor in the ARC nuclei [60,74]. However, testosterone effects in the AVPV nuclei are mediated by androgen receptor [60,74]. Other studies have shown that estrogens can inhibit KP expression in the ARC nuclei but stimulate KP expression in the AVPV nuclei [75,76] (Figure 3).

Activation of ERα can mediate either negative feedback in ARC nuclei or positive feedback in AVPV nuclei in female [74]. Active estrogen response elements have been identified in the proximal as well as distal enhancers loci of the KP gene [77,78]. Thus, ERα plays an essential role for proper regulation of GnRH and GPN secretion from the HP axis [74,79]. In ERα^KO^ mice and rats, high levels of GnRH and GPNs are detected despite high levels of sex steroid hormones due to the lack of negative feedback response [80,81]. In contrast, an attenuated preovulatory GPN surge is observed in ERβ^KO^ mice and rats due to decreased estrogen synthesis in the ovary [82,83]. In addition to the sex steroid hormone receptors, transcription factor NHIH2 plays an important role in sex-specific pubertal regulation of KP and neurokinin B gene expression [84]. Recently, another transcriptional regulator VAX1 has been found to regulate the sex-specific differential expression of KP gene in ARC and AVPV nuclei [85].

Moreover, gamma-amino butyric acid B (GABAB) receptor signaling can regulate *Kiss1* expression in both sexes [86]. Inhibition of GABAB signaling with a selective antagonist (CGP) reduced the expression of *Kiss1* and *Tac2* (neurokinin B) genes in the ARC nucleus [86]. CGP-treatment resulted in decreased FSH level and delayed the onset of puberty in male mice, whereas increased the number of atretic and decreased the number of ovulatory follicles in females [86]. Studies also suggest the importance of epigenetic mechanisms regulating hypothalamic KP expression [87]. We identified hat ERβ plays an essential role for expression of KPs in ovarian granulosa cells [43]. However, compared to estrogen regulation of KP expression in hypothalamic KP neurons, very little is known about the regulation of extrahypothalamic KPs.

**Figure 3 cells-11-01146-f003:**
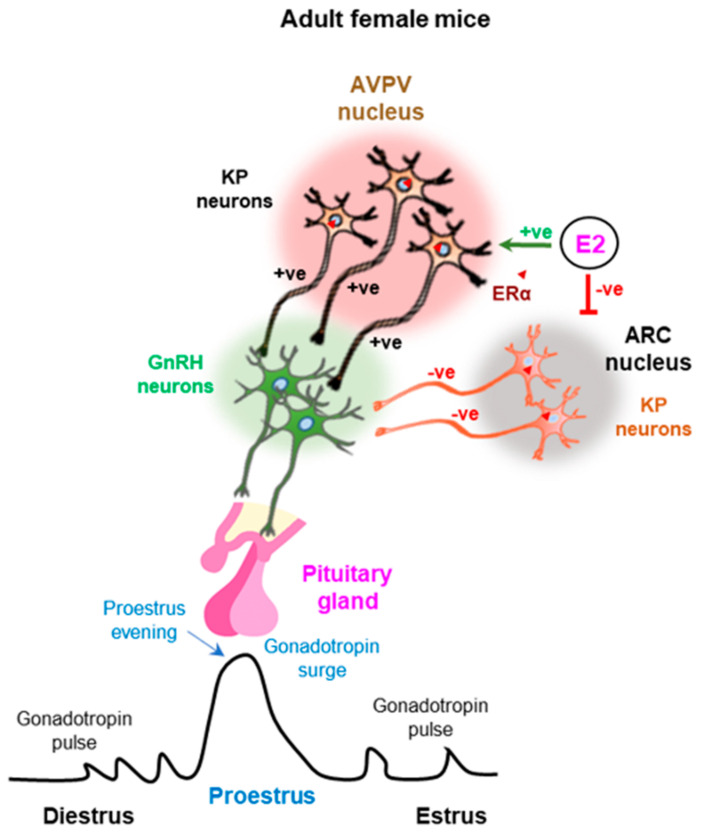
Estrogen feedback regulation of kisspeptin neurons in the arcuate and anteroventral periventricular nuclei. While estrogen negatively regulates the kisspeptin (KP) neurons in the arcuate (ARC) nuclei, positively regulates those in the anteroventral periventricular (AVPV) nuclei. ERα-mediated estrogen signaling in the KP neurons of ARC nuclei induces the pulsatile gonadotropin secretion during the estrus cycle. In addition, estrogen signaling in the KP neurons of AVPV nuclei induces preovulatory gonadotropin surge in proestrus evening. +ve, positive; −ve, negative.

## 2. Sexual Dimorphism in Hypothalamic Kisspeptin Expression

The onset of puberty, and reproductive functions are all dependent on the KP and KPR system [3,88]. Thus, it is imperative that a sexual dimorphism in KP and KPR expression leads to distinct pattern of GnRH release and GPN secretion in males and females.

### 2.1. Dimorphic Expression of Kisspeptins during Prenatal Development

In the adult ARC, sex-specific differences in KP expression are more subtle than in the AVPV/PeN within preoptic area (POA) [89,90]. Thus, the sexual dimorphism in KP neurons in the ARC are sometimes reported at the morphological [91] and functional levels [92]. KP neurons in ARC regulate the onset of puberty as well as the cyclic regulation of GnRH pulses in both sexes. These neurons also influence the AVPV/PeN regulated GPN surges in females [93,94,95]. KP expression in the ARC is turned on very early in the fetal brain [90,96]. At embryonic day 16.5 (E16.5), the number of KP neurons are similar in both male and female ARC, but the KP mRNA expression per cell is significantly higher in female [97,98]. A higher level of *Kiss1*, and *Esr1* but a lower level of *Ar* are detected in the females ARC nuclei [97,98]. A higher level of *Tac2* (neurokinin B) and *Tacr3* expression in these KP neurons suggest an increased neurokinin B signaling in the female ARC [97,98]. Such sexual dimorphism in the ARC nuclei at this developmental stage plays a key role in the specification sex-specific responses including the male-specific prenatal testosterone surge [90,96,97,98].

### 2.2. Dimorphic Expression of Kisspeptins during Postnatal Development

Hypothalamic expression of KP mRNAs show clear sexual dimorphism during postnatal development [99]. In the ARC nucleus, KP mRNAs is detectable starting at postnatal day (PND) 3, and gradually increases during further postnatal development [99] (Figure 4). The number of KP neurons are greater in neonatal females than that of the males, which are decreased in juvenile period but increased again at early adulthood [99]. The expression of KP mRNA is increased further during the onset of puberty in both sexes [99]. In contrast, KP mRNAs are detected in the male AVPV nuclei as early as on PND7, whereas it becomes detectable in females on PND21 [99].

Nevertheless, the number of KP neurons in AVPV nuclei increases in both sexes during the postnatal development [99] (Figure 5). These findings indicate that KP neurons appear earlier in ARC nucleus than those in AVPV and show a sex-specific difference in their numbers during postnatal development [99]. In addition to the sexually dimorphic KP neurons, the AVPV/PeN contains sexually dimorphic tyrosine hydroxylase (TH) neurons that synthesize catecholamines, specially dopamine [100].

Females possess more TH neurons than males [100]. The vast majority of KP neurons in the AVPV/PeN express *Th* mRNA and protein, suggesting that dopamine, like KP, may regulate GnRH secretion and participate in the positive feedback to estrogen. However, selective ablation *Th* expression in KP neurons failed to affect the onset of puberty, reproductive hormones, and fertility in both females and males [101].

### 2.3. Distribution of Disspeptin Neurons and Kisspeptin Receptors in Adult Hypothalamus

KPs are expressed in high levels in the brain, particularly in the hypothalamic region [102]. In addition to the hypothalamus, KPRs are expressed in the pons, midbrain, thalamus, hippocampus, amygdala, cortex, frontal cortex, and striatum [103]. Rodent KP neurons are located in two major brain regions: ARC and AVPV nuclei [104,105,106] (Figure 6). Distribution of KP neurons in rodent AVPV nuclei extends caudally to the PeN zone, together known as the rostral preoptic area of the third ventricle (RP3V). In contrast to rodents, primates and ruminants do not have a defined RP3V area [105]. KP neurons in the preoptic region in sheep, goats and primates are homologous to the RP3V in rodents [107,108], but the precise function of KP neurons differs slightly among different species [105]. The density of KP neurons and the innervation pattern differs in the ARC/VPN and AVPV/PeN nuclei between the males and females [104] (Figure 6). KP neurons are more abundant in female AVPV/PeN nuclei than that of male, and higher levels of KP mRNAs are detected in females [109]. In contrast to the AVPV/PeN nuclei, KP neurons in the ARC nuclei show a minimal degree of sexual dimorphism [110] (Figure 6).

### 2.4. Expression of Hypothalamic Kisspeptins during Aging

Aging is associated with a decline in the signaling activity of the HPG axis, resulting in hormonal abnormalities [64,111]. In ageing rats, the number of KNDy neurons are reduced, resulting in a reduced level of pulsatile secretion of GnRH [64]. Both mRNA and protein levels of KPs and KPR are significantly decreased in the hypothalamus of ageing rats [112]. However, changes in the numbers of KP neurons and level of KP expression in humans differs from their rodent counterparts [113]. During aging in humans, hypothalamic KP and NKB neurons increase in numbers, undergo hypertrophic changes, and show increased expression of both neuropeptides [114]. Although the changes occurs in both sexes, a much greater effect is observed in females [114]. In postmenopausal women, expression of KP increases in the infundibular nuclei [107]. These findings are supported by studies showing that removal of the sex steroids increases the expression of the KP gene [4,74,107,115]. Ovariectomized monkeys also displayed an increased number of KP neurons and KP neuron hypertrophy. Due to the lack of ovarian-derived steroid hormones in post-menopausal women, there is elevated level of KPs due to reduced negative feedback that leads to increased secretion of GnRH and GPNs [116]. Similarly, reduced serum levels of free testosterone in ageing males results in increased levels of GPNs [117]. The number of KP neurons were significantly increased in hypothalamic tissues of men older than 50 years, and was associated with an increased numbers of NKB positive neurons [118].

## 3. Kisspeptin Regulation of Gonadotropin Secretion

### 3.1. Kisspeptin Regulation of GnRH and Gonadotropins

KP is the major regulator of hypothalamic GnRH secretion from hypothalamic neurons. KP neurons are organized in apposition to GnRH neurons within the hypothalamus of both males and females [18,119,120,121]. KPs act on the KPR expressed by GnRH neurons and induces GnRH release [122,123,124,125,126,127]. GnRH is released into the median eminence, picked up by a capillary plexus and is delivered to the anterior pituitary. There it acts on GnRH receptors expressed by pituitary gonadotrophs and induces the secretion of the GPNs- FSH and LH. KPs from ARC nuclei generate pulsatile secretion of GnRH, and in turn GPNs from the pituitary gland in both sexes, which are essential for the onset of puberty, gonadal development, steroidogenesis, gametogenesis, and other reproductive functions [125,128]. In addition, women express significantly higher levels of KP in AVPV nuclei and PeN area, which is responsible for the induction of the preovulatory GnRH and GPN surge [109]. The preovulatory GPN surge in the females is required for final stages of follicle development, oocyte maturation, ovulation, and formation of corpora lutea [129,130]. Clinically, exogenous kisspeptin-54 administration induces LH secretion and egg maturation in women undergoing in vitro fertilization [131]. Administration of KP antiserum can block the estrogen-induced GnRH and GPN surge, which proves the essential role of KPs in AVPV nuclei for the surge induction [119,132]. Thus, hypothalamic KPs and KPR serve as key regulators of gonadal functions through GnRH and GPN secretion [129,130].

### 3.2. Gonadal Steroids Regulating GnRH and Gonadotropin Secretion

Gonadal steroid hormones regulate the expression of KPs, which is the key regulator of GnRH secretion. However, steroid hormones regulate the KP neurons in the ARC and the AVPV nuclei differentially; while the steroid hormones negatively regulate KP expression from ARC nuclei, they act as a positive regulator of KP expression in the AVPV nuclei and PeN area [93,133]. The density of KP neurons in AVPV and ARC nuclei, and the innervation of the KP neurons in hypothalamic paraventricular (PVN) nuclei differ between the males and females [104]. The numbers of KP neurons in AVPV nuclei is minimal in the males and KP expression for AVPV nuclei is predominantly observed in the females [104]. KP levels in the AVPV nuclei are higher than that of ARC nuclei and coincide with induction of the GPN surge from the HP axis [104] (Figure 3). Gonadal steroid hormones play a critical role in the sexual dimorphism in KP and GPN secretion. Sex-specific differences in the number of KP neurons in AVPV nuclei is dictated by gonadal hormones during the early neonatal development [134]. The importance of gonads in the establishment of sexual dimorphism in KP and GPN responses have been experimentally demonstrated in mice models [109]. When neonatal male rodents are castrated, they acquire the GnRH and GPN surge like that observed in females [109]. In contrast, treatment of female mice with androgen on the day of birth results in depletion of KP neurons in the AVPV region like that of the males [109]. Females exposed to testosterone also lose their ability to generate a GnRH and GPN surge [109].

### 3.3. Kisspeptin Regulation of Neonatal Testosterone Surge

A neonatal testosterone surge (NTS) occurs in males within a few hours after birth [135]. After this brief elevation, the testosterone level drops and remains low until the onset of puberty [135] (Figure 7). The NTS is partially responsible for establishing sexually dimorphic brain function that controls male reproductive physiology [136,137]. NTS also plays an important role in the development of male reproductive organs and reproductive function [138,139,140,141,142]. Remarkably, there is no equivalent neonatal estradiol or testosterone surge in females [143]. The transient activation of HPG axis in neonatal males results in a transient rise in serum testosterone levels [138,144,145], which is associated with elevated serum LH [146]. Although the mechanism underlying generation of NTS in males remain largely unclear, activation of GnRH neurons occurs 0–2 h after birth and correlates with the NTS occurring in mice [147].

GnRH receptor knock-out neonatal male mice lack an NTS, and subsequently develop a female-like brain (demasculinized) in adulthood [148]. Targeted depletion of KPR in GnRH neurons also leads to the lack of NTS and sexual differentiation of male brains [148,149]. KPs are a potent direct activator of GnRH neurons [147]. A population of KP neurons appear in the PeN area of male embryos between E19.5 and PND1. Transient activation of these KP neurons drive the neonatal GnRH surge [150] (Figure 7). These findings indicate that perinatal KP inputs to GnRH neurons are essential for the male-specific NTS driving the sexual differentiation of the male brain [109,147,150]. Furthermore, it is thought that NTS leads to depletion of KP neurons in the RP3V region of the brain to ultimately establish the sexual dimorphic features of hypothalamic KPs in male mice [18,109,149,150,151].

## 4. Physiological Functions of Hypothalamic Kisspeptins

During the onset of puberty, KPs increase the release of GnRH from the hypothalamus that induces GPN secretion from the pituitary gland [152]. GPNs regulate gonadal development and syntheses of gonadal hormones. Sex-specific differences in the distribution of KP neurons in hypothalamic nuclei and differential KP expression in response to gonadal hormones result in the sexual dimorphism in reproductive physiology, including onset of puberty, gonadal development and reproductive functions [110].

### 4.1. Kisspeptin Regulation of the Onset of Puberty

KP-signaling plays a vital role in controlling the onset of puberty in both male and female [153,154]. An increase in GnRH release from the hypothalamus triggers the onset of puberty. At puberty, an increased level of GnRH release and GPN secretion have been reported in many mammalian species including humans [155,156]. KPs expressed by the KP neurons in ARC nuclei are critical for generating the hypothalamic GnRH pulses, which is obligatory for the onset of puberty [157]. However, KPs from ARC nuclei do not dictate the timing of pubertal onset [157]. Although the number of KP neurons remain unchanged in the female ARC nuclei, KP mRNA levels are elevated about fourfold at PND26 in female rats [157,158]. In contrast, KP mRNA levels are significantly elevated at PND45 in the male rats, and the number of KP neurons increases throughout postnatal development [157,159]. Expression of KP mRNA is also significantly elevated in AVPV nuclei of both male and female rats during the onset of puberty [115]. It has been reported that pulsatile infusion of GnRH can induce precocious puberty in immature guinea pigs and monkeys [160,161]. An increased level of pulsatile GnRH release induces GPN secretion, which induces folliculogenesis and estradiol secretion in the females [162]. Whereas an increased level of FSH and elevated testicular testosterone production initiate spermatogenesis in the males [163].

### 4.2. Kisspeptin Regulation of Gonadal Development and Function

Defective KP-signaling disrupts gonadal development due to a lack of GnRH and GPN secretion [26]. A lack of KP or KPR functions also results in hypogonadotropic hypogonadism in men and women [26]. In mouse or rat models of KP or KPR mutants, development of the gonads adversely affected in both male and females [164]. KP neurons in the ARC nucleus regulate pulsatile secretion of GPNs, which are required gonadal development, steroidogenesis, and gametogenesis in both males and females. An increased level of pulsatile GnRH release induces GPN secretion leading to 1) ovarian development, folliculogenesis, and estradiol secretion in the females [165], and 2) increased testicular testosterone production and the initiation of spermatogenesis in the males [166]. In contrast to the KP neurons in ARC nuclei, those in AVPV nuclei contribute to the preovulatory GnRH and GPN surge in females which is absent in males. Development of ovarian follicles beyond the antral stage, oocyte maturation, and ovulation requires stimulation by the preovulatory GPN surge. Ovulation in most mammalian species requires a large surge of GnRH and GPNs in response to rising estrogen levels that induce KPs from the AVPV nuclei [167].

### 4.3. Nonreproductive Role of Hypothalamic Kisspeptins

KP expression in the ARC nuclei is regulated by metabolic cues including leptin and insulin [168]. Thus, metabolic alterations such as obesity and malnutrition, affect the KP expression and the production of GnRH and GPNs [169,170]. Recent studies also suggest that KP neurons in ARC nuclei regulate lactation [171], appetite and growth hormone release [172]. Findings in whole body knockout models of KPR (KPR^KO^) suggest that KP-signaling plays an important role in energy expenditure, food intake and body weight gain and this is sexually dimorphic [173,174]. KPR^KO^ female mice gained body weight and developed impaired glucose tolerance, but the KPR^KO^ male mice suffered loss of body weight due to a suppression of feeding [174,175]. Selective reintroduction of KPR into the GnRH neurons could partially rescue the metabolic phenotypes in both male and female mice, which suggests that hypothalamic KP signaling plays a role in the metabolic regulation [174]. PTEN is a critical regulator of metabolism, which is a major negative regulator of the PI3K/AKT pathway and plays an important role in both lipid and glucose metabolism [168]. Targeted deletion of *Pten* gene in KP neurons increased the activation of mTOR-signaling, the central regulator of metabolism, in both ARC and AVPV nuclei in female mice but not in males [168]. Nevertheless, the precise molecular mechanisms that integrate metabolic signals to KP neurons remain largely unclear [168].

## 5. The Biology of Extrahypothalamic Kisspeptins

### 5.1. Extrahypothalamic KPs inside Brain

Recent studies suggest that cells that express KP may be present in widespread areas of the brain outside the hypothalamus and limbic system [68,176]. KP promoter activity has been detected in widespread cortical areas including layers 5 and 6 of the neocortex, insular cortex, and piriform cortex, as well as in the lateral septum, and in the nucleus of the solitary tract in the brainstem [68,176] (Figure 8).

The localization of KPs in the circuitry that mediates pheromonal control of sexual and emotional behavior and neuroendocrine function [177,178] suggests that KP-signaling may regulate these functions [179]. Overall, the role of and function for KPs in the neocortex and other brain regions currently remains a mystery but indicates potential role outside of the neuroendocrine regulation of reproduction [105].

### 5.2. Extrahypothalamic Kisspeptins in Gonadal Functions

Reproductive function of KP-signaling within the HP axis is well recognized. However, KP and KPR are also expressed in the testis and ovary [28]. While granulosa cells in ovarian follicles express KPs, oocytes are the major site of KPR expression (Figure 9). It was initially suggested that the KP and KPR system in the ovary and the uterus is independent of the HPG axis; however, recent studies have revealed that GPN-stimulation is essential for the induction of KP expression in follicular granulosa cells [180]. Rat ovaries showed an increased expression of KPs during the preovulatory period [28]. In the ovaries, KP-signaling likely plays a role in steroidogenesis, follicle maturation, oocyte survival, and ovulation [181]. It was shown that KPR haplo-insufficient mice develop an excessive age-related loss of ovarian reserve and premature ovarian senescence, and this was independent of circulating GPN levels [181]. Loss of KPR prevented follicle maturation and ovulation and could not be rescued by administration of exogenous GPNs [182]. Studies suggested that KPs and neurotropins act together to promote the oocyte survival in antral follicles [183]. Recent studies have also shown that KP stimulation can improve in vitro maturation of oocytes [184,185]. KP treatment triggered intracellular Ca2+ release in rat oocytes and was associated with ERK1/2 activation [186].

Similar to that of intraovarian KP signaling, KP and KPR are also detected in the sperm, and they modulate sperm functions [187,188,189,190,191] (Figure 9). KPs are not detected in spermatids but KPR is expressed in the acrosomal region of spermatids and mature spermatozoa [192]. While KPR is expressed in both Leydig cells and Sertoli cells [192], expression of KP was detected only in Leydig cells [39]. KPs can impact male fertility by improving male gametogenesis [193]. It was shown that expression of KPR into GnRH neurons of KPR^KO^ mice was unable to compensate for the defective testicular function indicating a direct role of KP-signaling in testis [194]. Treatment with KPs improves human sperm motility and induces their transient hyperactivation, which could be blocked by a KPR antagonist KP-234. KP treatment triggers intracellular Ca2+ release in sperms and that can be inhibited by KP-234 resulting in reduced in vitro fertilization [192].

### 5.3. Kisspeptin-Signaling in Embryo Implantation and Placentation

KP and KPR are expressed in rodent as well as human uterus [195,196]. Heterozygous KP mutant embryos failed to implant into KP^KO^ mouse uteri but implanted in the wildtype [197]. The failure of implantation could not be rescued by exogenous GPNs, estradiol, and progesterone but leukemia inhibitory factor (LIF) was able to help implantation [197]. KP promotes embryo adhesion to endometrium by up-regulating adhesion molecules [197]. KPs also increase the expression of LIF and improve decidualization [197]. Like many other endocrine tissues, high levels of KPs and KPR are expressed in the placenta [44,46]. The placenta releases hormones into both maternal and fetal circulations and support fetal growth and modulate maternal metabolic adaptation to pregnancy [198]. KP-signaling plays a vital role in regulating the placental functions [46]. A higher level of KPR expression in the first trimester placenta suggest that KP-signaling may be more important during early pregnancy [199,200]. However, it remains undecided whether placenta-derived KPs are sufficient for the enormous quantity of circulating KPs during pregnancy.

### 5.4. Extrahypothalamic Kisspeptins in Metabolism

KPs and KPR are expressed in liver, pancreas, and adipose tissues that are key peripheral sites for metabolic regulation [201]. There have been growing evidence that KPs regulate metabolism directly through the presence of KPR in liver, pancreas, and adipose tissues, and indirectly by stimulating gonadal hormones [174]. KPR^KO^ female mice displayed an increased body weight and adiposity associated with increased serum leptin levels, and impaired glucose tolerance [174] (Figure 10). In contrast, KPR^KO^ male mice had decreased feeding and reduced body weight but no impairment in glucose tolerance (Figure 10).

KP and KPR have been detected in both α and β cells of mouse and human pancreatic islets [202]. Administration of exogenous KP-54 increased glucose-induced insulin secretion [202]. While intravenous administration of exogenous KP-10 increased circulating insulin levels, administration in ventricular space did not, suggesting a peripheral site of KP action [203]. It has been suggested that islet-derived KPs act in an autocrine manner and potentiate glucose-stimulated insulin secretion [203].

Data from in KPR^KO^ mice suggest that KP-signaling may impact energy expenditure, food intake and body weight in a sexually dimorphic manner [173,174]. The metabolic and diabetic phenotypes in KPR^KO^ mice could be due to reduced pancreatic β-cell function and the absence of peripheral KP-signaling [202,203]. It can also be mediated by a direction action of KPs on the KPR in the brain, pancreas, and brown adipose tissues or via GPN induction [173,175]. Understanding the sexually dimorphic phenotype of KPR^KO^ females may also give insight into polycystic ovarian syndrome, a common reproductive and metabolic syndrome seen in 10% of all reproductive aged women [204]. However, further studies are required to understand the sexually dimorphic nature of metabolic phenotype [202,203].

## 6. Conclusions

Sexual dimorphism between males and females is most prominent in their reproductive organs and functions. As gonadal development and functions are dependent on KP-signaling, expression and function of KP and KPR show a remarkable sexual dimorphism. KP-signaling serves to link the gonadal steroid hormones and the GnRH neurons. Gonadal steroid hormones are the upstream regulators of KP expression in hypothalamic neurons. Gonadal hormones and their production are different in males and females, resulting in an obvious sexual dimorphism in KP-expression. Gonadal hormones mediate the negative feedback regulation of ARC nuclei resulting in pulsatile GnRH and GPN secretion, which is essential for gonadal development, onset of puberty, steroidogenesis, and gametogenesis in both sexes. While spermatogenesis is a continued process, follicle development, oocyte maturation and ovulation are GPN-dependent but cyclic event. Linked to these dimorphic gonadal functions during gametogenesis, a sexual dimorphism in KP-signaling exists in the AVPV/PeN nuclei that leads to the preovulatory GPN surge. Preovulatory KP response and the GPN surge is required for the maturation of ovarian follicles and induction of ovulation. Subsequent fertilization of oocytes, implantation of embryos, pregnancy, and embryo development also occur only in the females. KP-signaling is important for all these dimorphic physiological functions and both hypothalamic and extrahypothalamic KP-signaling are shown to play key regulatory roles. Gonadal steroids and the biology of differential reproductive functions are also linked to sexually dimorphic metabolic adaptations to KP signaling. A huge increase in KP production during pregnancy, which is likely placental and/or hepatic origin, may be required for maternal metabolic adaptation. Gonadal KPs and KPR have been found essential for testicular function; moreover, ovarian-derived KPs also play a role in oocyte maturation and induction of ovulation. Administration of exogenous KPs in peripheral sites elicits GPN secretion in the HP axis; however, it remains undetermined whether extrahypothalamic KPs of peripheral tissue origin play a role in the regulation of GnRH and GPN secretion.

## Figures and Tables

**Figure 1 cells-11-01146-f001:**
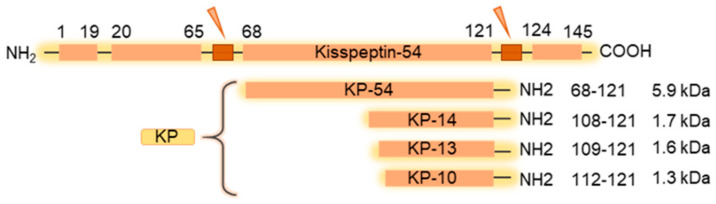
Expression and processing of pre-pro-kisspeptin. Kisspeptins (KPs) are encoded as a common precursor. The pre-pro-kisspeptin is a 145 amino acid peptide, with a 19-amino acid signal peptide and a main 54-amino acid region (52 amino acid in in rodents), kisspeptin-54/52 (KP-54/52). KP-54/52 is further processed into peptides of lower molecular weight: KP-14, KP13, and KP-10. All KPs contain the RF-amide (in primates) or RG-amide (in rodents) motif in the C-terminus that can bind and activate the KP receptor.

**Figure 2 cells-11-01146-f002:**
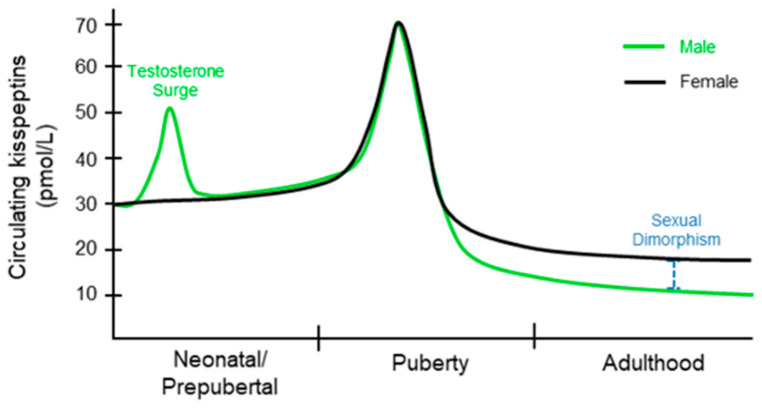
Circulating kisspeptin levels in men and women. Kisspeptins (KPs) are detected in the human plasma. Sex specific events, such as neonatal testosterone surge in males and pregnancy in females, can affect the plasma KP levels. KP levels also change with increasing age but exhibit significant sexual dimorphism after puberty. In both men and women, the highest KP levels are detected during puberty and thereafter, plasma KP levels decrease with aging.

**Figure 4 cells-11-01146-f004:**
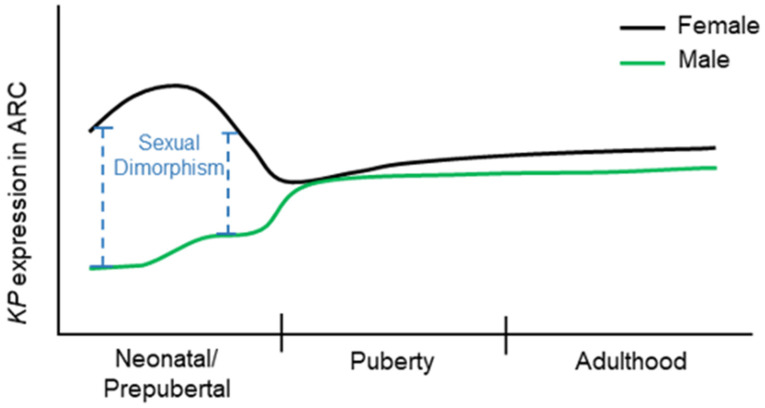
Kisspeptin expression in arcuate nuclei. The changes in kisspeptin (KP) expression in females (black line) and males (green line) were traced through the neonatal/prepubertal periods, the pubertal period, and the adulthood in previous studies. There is a high degree of sexual dimorphism in KP expression in the arcuate (ARC) nuclei during neonatal and prepubertal period. The sex-specific dimorphic expression of KPs in ARC nuclei sustains until the onset of puberty when the levels of KP expression become similar.

**Figure 5 cells-11-01146-f005:**
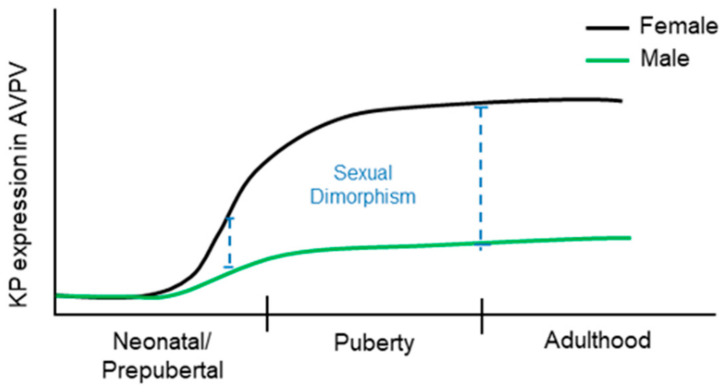
Kisspeptin expression in anteroventral periventricular and preoptic nuclei. The changes in kisspeptin (KP) expression in females (black line) and males (green line) were traced through the neonatal/prepubertal periods, the pubertal period, and the adulthood in previous studies. Until the second postnatal week, the expression of KP in the anteroventral periventricular and periventricular preoptic nucleus (AVPV/PeN) was not detectable. After KP expression becomes detectable in this region, a distinct sexual dimorphism persists throughout the postnatal life.

**Figure 6 cells-11-01146-f006:**
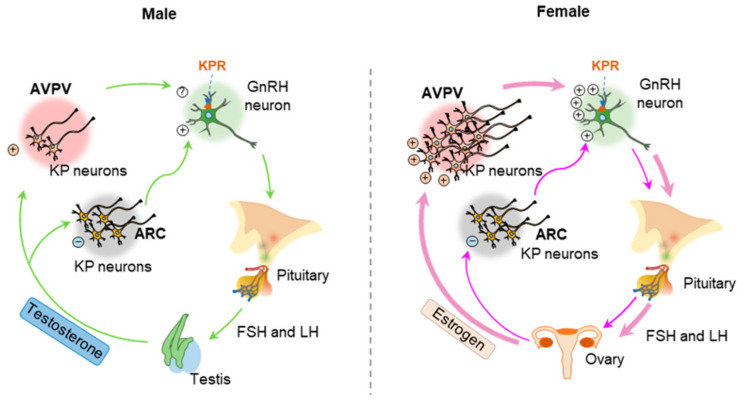
Sexual dimorphism in hypothalamic kisspeptins. Kisspeptin (KP) neurons are located in two major regions: the arcuate (ARC) nuclei, and the anteroventral periventricular and periventricular preoptic nuclei (AVPV/PEN). KPR are expressed on the GnRH neurons. KPs expressed in hypothalamic nuclei are crucial for the onset of puberty, gonadal development, and maintenance of reproductive functions. Sex-specific differences in the numbers of KP neurons and the KP expression levels exist in the hypothalamus. KP neurons in ARC nuclei generate pulsatile GnRH secretion, while those in the AVPV nuclei induce the preovulatory gonadotropin surge (thick line). GnRH neurons, found near the pituitary gland, is innervated by the KP neurons of ARC and AVPV origins. While the KP expression in the ARC nuclei is similar between males and females, KP expression is highly dimorphic in the AVPV nuclei due to greater numbers of KP neurons in females. Thin lines stand for lower levels and thick lines stand for higher levels of expression.

**Figure 7 cells-11-01146-f007:**
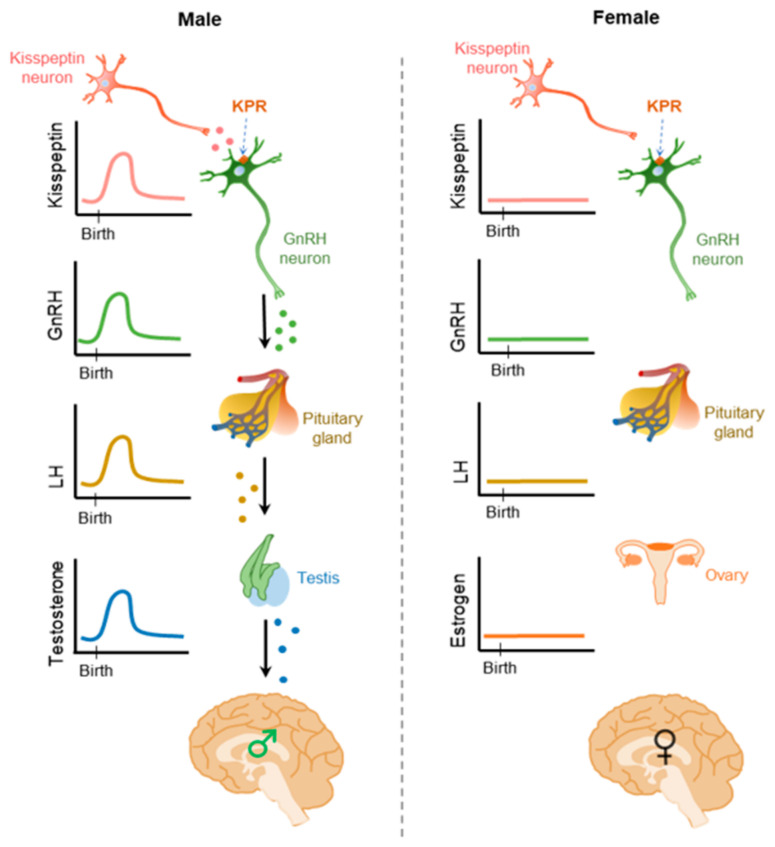
Neonatal testosterone surge in males. In the males, there is a transient increase of testosterone level soon after the birth. This event, also known as “mini-puberty” is driven by male-specific neonatal kisspeptin (KP) neurons and results in the activation of the hypothalamic-pituitary axis. There is no parallel phenomenon in the females. This neonatal testosterone surge (NTS) is critical for the establishment sexual dimorphism in the AVPV/ PeN nuclei depleting the KP neurons in males. The NTS is also important for the normal development of male reproductive system.

**Figure 8 cells-11-01146-f008:**
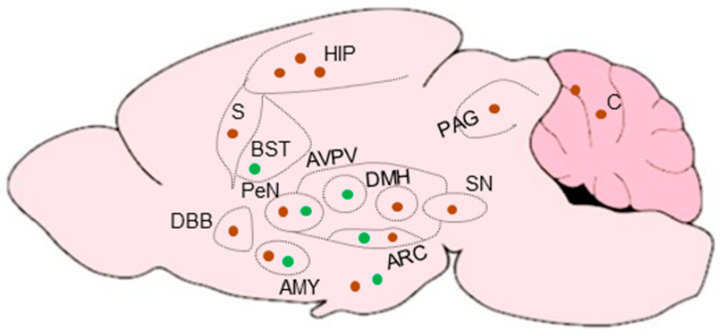
Hypothalamic and extrahypothalamic expression of kisspeptins and kisspeptin receptor in the brain. Anatomical distribution of kisspeptin (KP) mRNA (green dots) and KP receptor (KPR) mRNA (brown dots) in different brain areas of rodents. KP mRNA expression is mostly detected in the hypothalamus (ARC, AVPV, and PeN). Presence of KP mRNAs has also been reported in the amygdala (AMY) and bed nucleus of stria terminalis (BST). KPR mRNA has been detected in different forebrain areas, including the diagonal band of Broca (DBB), and septum (S).

**Figure 9 cells-11-01146-f009:**
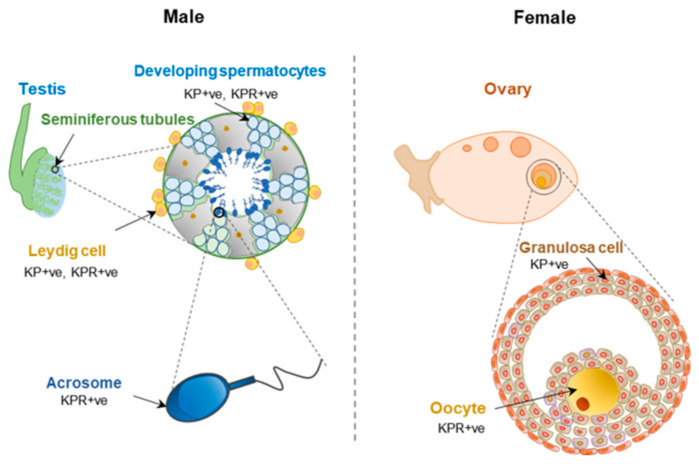
Extrahypothalamic kisspeptin and kisspeptin receptor in gonads. Kisspeptins (KPs) and KP receptor (KPR) are expressed in the sperm. KPs are not detected in spermatids but KPR is expressed in acrosomal region of spermatids and mature spermatozoa. KPs and KPR are also detected in the ovaries. While KPs are predominantly expressed by the granulosa cells, KPR is expressed in the oocytes. KPR+ve, stands for kisspeptin receptor positive.

**Figure 10 cells-11-01146-f010:**
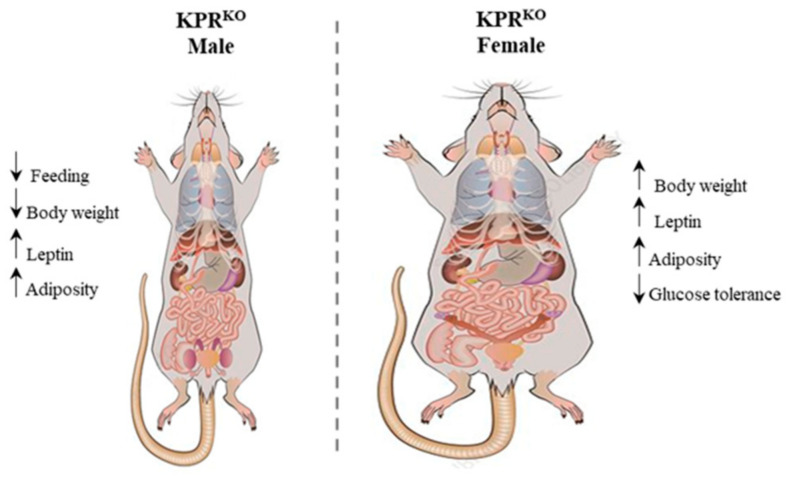
Role of kisspeptin signaling in metabolism. Kisspeptins (KP) and KP receptor (KPR) are present in liver, pancreas, fat, and gonad tissues. KPR^KO^ male mice suffered from with feeding suppression and decreased body weight. They had increased adiposity but did not develop glucose intolerance. In contrast, KPR^KO^ female mice gained bodyweight, higher leptin levels, marked adiposity with impaired glucose intolerance. The findings in KPR^KO^ mice suggest that KP-signaling impacts energy expenditure, food intake and body weight gain in a sexually dimorphic manner.

## Data Availability

Not applicable.

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
