# Peer review of "Sexual Dimorphism in Kisspeptin Signaling"

_cells, 2022, doi:10.3390/cells11071146_

Round 1

Reviewer 1 Report

General comments

The topic of the manuscript is of great relevance, it is well written, it is easy to read and really very interesting.

I consider that it is publishable after the application of the corrections.

The image showed in figure 2 does not coincide with the text. Circulating KPs levels in children are higher than that of adults… and so on

It would be appropriate to add units of circulating kisspeptins on axis.

Line 138 – I would add that it may also be related to hormonal changes as sex hormones have been shown to influence kiss levels

Figure 3: Add in the title that it is a scheme of females.

Add + and – signs on graph to explain stimulatory and inhibitory effectsAgregar la kP y sus efectos sobre gnrh neurons

Add the effects of testosterone mentioned in the text

Line 208 It has “been” demonstrated

Figure 6 sexual dimorphism in hypothalamic kisspeptin neurons: Explain in the text the meaning of GPR54

Explain the meaning of thick and thin pink lines and the blue dashed line334 are inervated

Figura 7

According to line 223, the number of kp neurons in neonatal females is greater than in males. However, in figure 7 they are not even drawn for females, so it gives the idea that there are none or that it is greater in males.

Figure 9: What does KPR+ve mean?

The following points explain the same thing. Unify them.

5.4. Extrahypothalamic kisspeptins in metabolism. 4.3. Nonreproductive role of hypothalamic kisspeptins

Line 656

Pre-ovulatory KP response and the GPN surge are female-specific phenomena required for the maturation of ovarian follicles and induction of ovulation.

Line 658

Fertilization of oocytes, implantation of embryos, pregnancy, and embryo development also occur only in the females.

In the conclusion it would be important to include a line about the role of KP in metabolism.

Author Response

Authors response to the reviewer’s comments

We have addressed the reviewer’s concern and revised our manuscript according to the reviewer’s suggestions. We hope that the current version of the revised manuscript is suitable for publication in Cells.

Query 1: Comment on the authors' response to the query #1.

The topic of the manuscript is of great relevance, it is well written, it is easy to read and really very interesting. I consider that it is publishable after the application of the corrections.

Response 1: We appreciate the reviewer’s comments.

Query 2: The image showed in Figure 2 does not coincide with the text. Circulating KPs levels in children are higher than that of adults… and so on. It would be appropriate to add units of circulating kisspeptins on axis.

Response 2: We have corrected the error. We have also indicated the kisspeptin levels in pmol/L units.

Query 3: Line 138– I would add that it may also be related to hormonal changes as sex hormones have been shown to influence kiss levels.

Response 3: According to the reviewer’s suggestion, we have added the line “it may also be related to hormonal changes as sex hormones have been shown to influence KP levels”.

Query 4: Figure 3: Add in the title that it is a scheme of females. Add + and – signs on graph to explain stimulatory and inhibitory effects. Agregar la kPy sus efectos sobre gnrh neurons. Add the effects of testosterone mentioned in the text.

Response 4: We have added + and – signs on graph to explain stimulatory and inhibitory effects of estrogen (E2). We have also added + and – signs on the GnRH neurons.

According to the reviewer’s suggestion, we have mentioned testosterone regulation of KP neurons in the male ARC and AVPV/PeN regions in hypothalamus.

Query 5. Line 208 It has “been” demonstrated

Response 5. That line has been rewritten correctly and replaced.

Query 6: Figure 6 sexual dimorphism in hypothalamic kisspeptin neurons: Explain in the text the meaning of GPR54. Explain the meaning of thick and thin pink lines and the blue dashed line334 are innervated

Response 6: We have replaced the GPR54 by KPR, which stands for kisspeptin receptor. We have also explained the meaning of thin and thick lines; “thin lines stand for lower levels and thick lines stand for higher levels of expression”.

Query 7: Figura 7. According to line 223, the number of kp neurons in neonatal females is greater than in males. However, in figure 7 they are not even drawn for females, so it gives the idea that there are none or that it is greater in males.

Response 7: According to the reviewer’s suggestion, we have corrected Figure 7. “A population of KP neurons appear in the PeN area of male embryos between E19.5 and PND1. Transient activation of these KP neurons drive the neonatal GnRH surge (158) (Figure 7) (Line 460-462). Although KP neurons may be present in the male at this stage, those are reduced or eliminated by the neonatal GnRH surge.

Query 8: Figure 9: What does KPR+ve mean?

Response 8: +ve, stands for positive. We have explained it in the Figure legend.

Query 9: The following points explain the same thing. Unify them.

5.4. Extrahypothalamic kisspeptins in metabolism. 4.3. Nonreproductive role of hypothalamic kisspeptins

Response 9: We agree with the reviewer that both section 4.3 and section 5.4 are on kisspeptin regulation of metabolism. However, if we combine these two sections, then we will need to make two subtitles under that section: one as ‘hypothalamic’ and another as ‘extrahypothalamic’ metabolism. Therefore, we prefer to keep the metabolism under two sections: one as hypothalamic function (4.3) and another as extrahypothalamic function (5.4).

Query 10: The following points explain the same thing. Unify them.

Line 656, Pre-ovulatory KP response and the GPN surge are female-specific phenomena required for the maturation of ovarian follicles and induction of ovulation.

Line 658, Fertilization of oocytes, implantation of embryos, pregnancy, and embryo development also occur only in females.

Response 10: We have made the suggested correction. “Preovulatory KP response and the GPN surge is required for the maturation of ovarian follicles and induction of ovulation. Subsequent fertilization of oocytes, implantation of embryos, pregnancy, and embryo development also occur only in the females”.

Query 11: In the conclusion it would be important to include a line about the role of KP in metabolism.

Response 11: We have made the suggested correction. “Gonadal steroids and the biology of differential reproductive functions are also linked to sexually dimorphic metabolic adaptations to KP signaling”.

Reviewer 2 Report

The authors present a comprehensive review of the biology of kisspeptin. There are numerous reviews on the topic and new one it seems almost monthly. The authors have worked to distinguish this review by including additional focus on extrahypothalamic actions of kisspeptin. This focuses primarily on gonadal function but also includes a portion on metabolic functions. These wide-ranging functions (e.g., metabolism) vary greatly across species so it is not entirely clear if these biological functions are completely conserved across species. The review focuses mainly on mice with some rat and primate references included. That is fine as these are the primary lab species and where the bulk of the scientific literature is focused. Including all the other species in which kisspeptin has been tested would be unnecessary. A possible caveat here might be pigs. Kisspeptin receptor KO pigs do not show the changes in body weight that kisspeptin receptor KO mice do. These publications are in abstract and proceedings form and not identifiable through PubMed searches. I don’t think it is absolutely necessary, but from a comparative biology perspective it might be interesting. I am just not convinced the putative metabolic effects of kisspeptins may be as well conserved across species as are the reproductive effects. Nonetheless, the authors did not go too far off the trail in the metabolic section, which is probably good. 

The authors have highlighted differences in kisspeptin biology between males and females. This seems prudent as there are fewer reviews that illustrate this. The figures were useful and appropriately illustrative. In general, the manuscript was well written.

Specific comment:
159 in the sentence beginning “KP and KPR are also expressed…”; the words “but in Sertoli” indicates some thing is missing in this sentence.

Author Response

Authors response to the reviewer’s comments

Query 1: The authors present a comprehensive review of the biology of kisspeptin. There are numerous reviews on the topic and new one it seems almost monthly. The authors have worked to distinguish this review by including additional focus on extrahypothalamic actions of kisspeptin. This focuses primarily on gonadal function but also includes a portion on metabolic functions. These wide-ranging functions (e.g., metabolism) vary greatly across species so it is not entirely clear if these biological functions are completely conserved across species. The review focuses mainly on mice with some rat and primate references included. That is fine as these are the primary lab species and where the bulk of the scientific literature is focused. Including all the other species in which kisspeptin has been tested would be unnecessary. A possible caveat here might be pigs. Kisspeptin receptor KO pigs do not show the changes in body weight that kisspeptin receptor KO mice do. These publications are in abstract and proceedings form and not identifiable through PubMed searches. I don’t think it is absolutely necessary, but from a comparative biology perspective it might be interesting. I am just not convinced the putative metabolic effects of kisspeptins may be as well conserved across species as are the reproductive effects. Nonetheless, the authors did not go too far off the trail in the metabolic section, which is probably good. The authors have highlighted differences in kisspeptin biology between males and females. This seems prudent as there are fewer reviews that illustrate this. The figures were useful and appropriately illustrative. In general, the manuscript was well written.

Response 1: We agree with the reviewer’s comments. There is likely to be some difference on any topic or mechanisms we examine across species. The primary focus  of this review was to highlight the difference in kisspeptin signaling between male and female, and hypothalamic versus extrahypothalamic.  As our review have been limited to human and rodents (rat and mouse), we have mentioned this limitation in our ‘Abstract’ section.

Query 2.Specific comment:
159 in the sentence beginning “KP and KPR are also expressed…”; the words “but in Sertoli” indicates something is missing in this sentence.

Response 2. We have corrected the statement. “While KPR is expressed in both Leydig cells and Sertoli cells (201), expression of KP was detected only in Leydig cells (39)”.
